# Disc Diffusion and ComASP^®^ Cefiderocol Microdilution Panel to Overcome the Challenge of Cefiderocol Susceptibility Testing in Clinical Laboratory Routine

**DOI:** 10.3390/antibiotics12030604

**Published:** 2023-03-17

**Authors:** Gabriele Bianco, Matteo Boattini, Sara Comini, Giuliana Banche, Rossana Cavallo, Cristina Costa

**Affiliations:** 1Microbiology and Virology Unit, University Hospital Città della Salute e della Scienza di Torino, 10126 Turin, Italy; 2Department of Public Health and Paediatrics, University of Torino, 10124 Turin, Italy; 3Doctoral Programme of the Lisbon Academic Medical Centre—Ph.D. CAML, 1169-056 Lisbon, Portugal

**Keywords:** cefiderocol resistance, disc diffusion, ComASP® cefiderocol, ID-CAMHB, susceptibility testing

## Abstract

Cefiderocol susceptibility testing represents a major challenge for clinical microbiology. Although disc diffusion showed robustness to test cefiderocol susceptibility, large areas of technical uncertainty (ATU) are reported by current EUCAST breakpoints. Herein, we evaluated the in vitro activity of cefiderocol on a collection of 286 difficult-to-treat Gram-negative isolates using disc diffusion and ComASP^®^ cefiderocol microdilution panel. Broth microdilution (BMD) in iron-depleted Mueller–Hinton broth was used as reference method. Following the EUCAST guidelines, disc diffusion allowed to determine cefiderocol susceptibility (susceptible or resistant) in 78.6%, 88.1%, 85.4% and 100% of Enterobacterales, *P. aeruginosa*, *A. baumannii* and *S. maltophilia* isolates tested, respectively. ComASP^®^ cefiderocol panel showed 94% and 84% of overall categorical agreement and essential agreement. Only one very major error and two major errors were observed, for MIC values nearly close to the resistance breakpoint (2 mg/L). Overall, 20.5% of the carbapenemase-producing Enterobacterales that achieved ATU results by the disc diffusion method tested resistant by both ComASP^®^ panel and reference BMD. Conversely, all VIM-producing *P. aeruginosa* showed MIC values in the susceptible range (≤2 mg/L). Lastly, only six out of seven (85.7%) *A. baumannii* isolates showing inhibition zones <17 mm tested resistant by both ComASP^®^ panel and the reference BMD suggesting that inhibition zone <17 mm are not unequivocally suggestive of resistance. Our results, although obtained on a limited number of isolates, suggest that the combination of disc diffusion with a ComASP^®^ cefiderocol microdilution panel could be a viable solution to overcome the challenge of cefiderocol susceptibility testing in routine microbiology laboratories.

## 1. Introduction

The spread of multidrug-resistant (MDR) bacteria is a major public health threat. Carbapenemase-resistant Gram-negative bacteria, including Enterobacterales, *Pseudomonas aeruginosa* and *Acinetobacter baumannii* are considered superbugs in health care settings since they exhibit a multiresistance phenotype toward almost all commonly used classes of antibiotics [1,2]. They have been then recognized as high-priority pathogens for which development of new drugs are urgently needed by the World Health Organization [3]. In recent years, several antimicrobials have been approved for clinical use, including new β-lactam/β-lactamase inhibitor combinations, e.g., ceftolozane-tazobactam, ceftazidime-avibactam, meropenem-vaborbactam and imipenem-relebactam [4,5].

Knowledge of the antimicrobial susceptibility and local prevalence of specific carbapenemase enzymes is of paramount importance when selecting new β-lactam/β-lactamase inhibitor combinations, as not all β-lactamase inhibitors have activity against all classes of enzymes [6,7]. Avibactam, vaborbactam and relebactam have activity against class A β-lactamases but not against metallo-β-lactamase producers and OXA-23-like carbapenemase. Furthermore, only avibactam has activity on OXA-48-like carbapenemase [8].

Cefiderocol is a novel siderophore cephalosporin approved for the treatment of a broad spectrum of MDR Gram-negative pathogens, including metallo-β-lactamase producers and OXA-23-like-producing *A. baumannii*. It is structurally similar to cefepime (pyrrolidium group on the C-3 side chain) and ceftazidime (carboxypropyl–oxymino group on the C-7 side chain), characteristics that enhance hydrolytic stability against β-lactamases and transport across the bacterial outer membrane [9]. Moreover, binding to extracellular free ferric ions allows cefiderocol to be transported across the outer membrane via the iron transport system of Gram-negative organisms, overcoming resistance mechanisms such as efflux pump upregulation and porin channel mutations [9]. Cefiderocol has been evaluated in large international surveillance studies, revealing promising activity against carbapenem-resistant Gram-negative isolates [10]. However, reports on cefiderocol resistance are steadily increasing [11,12,13].

Based on the mechanism of cefiderocol, mutations affecting the iron transporter systems are associated with clinical resistance. Mutations in piuD and pirR, pirA and piuA, and cirA have been identified in *P. aeruginosa*, *A. baumannii* and *K. pneumoniae* clinical isolates, respectively [14]. However, the role of several β-lactamases in reducing susceptibility or inducing resistance to cefiderocol has been supported by several recent reports [14,15,16]. In this context, in vitro antimicrobial susceptibility testing is essential to enable the appropriate use of this last-resort drug. MIC determination by broth microdilution (BMD) represents the gold standard for cefiderocol susceptibility evaluation. However, it requires the use of cation-adjusted iron-depleted Mueller–Hinton medium (ID-CAMHB), whose preparation is time-consuming and difficult to implement in routine clinical microbiology laboratories. Recently, a CE-IVD BMD plate using a normal Mueller–Hinton broth was proposed, but it was subsequently withdrawn from the market due to lack of accuracy (https://www.eucast.org/ast-of-bacteria/warnings, accessed on 5 January 2023). Disc diffusion was shown to be robust and applicable to test cefiderocol susceptibility in Gram-negative bacteria [17] and was proposed by EUCAST guidelines as a first-line method (https://www.eucast.org/ast-of-bacteria/warnings, accessed on 5 January 2023) in routine testing. However, large areas of technical uncertainty (ATU) are reported by current EUCAST breakpoint tables (e.g., 18–22 mm for Enterobacterales and 14–22 mm for *Pseudomonas* spp.). Moreover, EUCAST MIC and inhibition zone diameter breakpoints for *Acinetobacter* spp. and *Stenotrophomonas malthophilia* are currently not available. A note indicates that inhibition zone diameters ≥17 mm and ≥20 mm for the cefiderocol 30 µg disc correspond to MIC values below the PK-PD breakpoint of susceptibility (≤2 mg/L) for *Acinetobacter* spp. and *S. malthophilia*, respectively. Inside the ATU, and as long as there is no alternative method to resolve interpretative uncertainties (e.g., routine laboratory MIC testing or support by a reference laboratory), EUCAST recommends ignoring the ATU and interpreting the inhibition zone diameter according to the table breakpoints. Therefore, alternative methods feasible in routine laboratories are urgently needed to resolve disc diffusion interpretative uncertainties.

The purpose of this study was to assess cefiderocol susceptibility on MDR Gram-negative isolates collected in Italian hospitals during a four-year period. For this aim, a diagnostic testing algorithm based on disc diffusion and the novel ComASP^®^ cefiderocol microdilution panel (Liofilchem^®^, Roseto degli Abruzzi, Italy) was implemented and evaluated.

## 2. Results

Overall, bacterial isolates included in the study were carbapenemase-producing Enterobacterales (*n* = 178), carbapenem-resistant *P. aeruginosa* (*n* = 42), carbapenem-resistant *A. baumannii* (*n* = 48) and *S. maltophilia* (*n* = 18). Enterobacterales isolates included 114 KPC producers (*K. pneumoniae n* = 110; *E.coli*, *n* = 4) of which, 44 showed ceftazidime/avibactam resistance, 19 NDM producers (*K. pneumoniae n* = 18; *E. coli*, *n* = 1), 31 VIM producers (Enterobacter cloacae, *n* = 15; *E. coli*, *n* = 5; *Citrobacter freundi*, *n* = 5; *Morganella morganii*, *n* = 2; *Proteus mirabilis*, *n* = 2; *Klebsiella aerogenes*, *n* = 1; *Serratia marcescens*, *n* = 1), 10 OXA-48-like producers (*K. pneumoniae*, *n* = 7; *E. coli*, *n* = 3) and KPC/VIM co-producers (*K. pneumoniae*, *n* = 4). In total, 10 out of the 42 carbapenem-resistant *P. aeruginosa* isolates harbored *bla*_VIM_, 46 out of the 48 *A. baumannii* harbored OXA-23 carbapenemase and 2 out of the 48 *A. baumannii* were OXA-23/NDM co-producers. Among Enterobacterales, disc diffusion achieved interpretable results in 140 out of the 178 isolates (78.6%). ATU results (*n* = 38) were mainly observed in ceftazidime/avibactam-resistant KPC (19 out of 44, 43.2%) and NDM producers (6 out of 19, 31.6%) (Table 1). Rates of resistance to cefiderocol by disc diffusion were higher in ceftazidime/avibactam-resistant KPC producers (47.7%) and in NDM producers (31.6%). Cefiderocol resistance rate according to disc diffusion was significantly higher in the ceftazidime/avibactam-resistant KPC-producer subset than in the susceptible ones (47.7% vs. 2.8%, *p* < 0.001) (Table 1). Likewise, a significant discrepancy in the pattern of inhibition zone diameter distributions was observed in the two subsets (*p* < 0.001). Five (11.9%) *P. aeruginosa* isolates exhibited inhibition zones inside the ATU, and all the remaining isolates tested susceptible. Inhibition zones <17 mm and ≥17 mm were observed in 7 (14.6%) and 41 (85.4%) *A. baumannii* isolates, respectively. All *S. maltophilia* isolates were found to be susceptible to cefiderocol, exhibiting inhibition zone diameters ≥28 mm.

MIC determinations obtained by ComASP^®^ cefiderocol microdilution panel and reference BMD on Enterobacterales, *P. aeruginosa* and *A. baumannii* isolates that achieved uninterpretable results by disc diffusion are reported in Table 2. Overall, 9 out of the 38 (23.7%) Enterobacterales that showed ATU results by disc diffusion tested resistant by reference BMD (MICs range: 4–16 mg/L). All *P. aeruginosa* tested susceptible (MICs range: 0.06–2 mg/L) and six out of the seven (85.7%) *A. baumannii* isolates tested resistant (MICs range: 4–16 mg/L). The ComASP^®^ cefiderocol microdilution panel showed 94% and 84% overall categorical agreement (CA) and essential agreement (EA), respectively. One very major error (VME) was obtained in a ceftazidime/avibactam-resistant *K. pneumoniae* exhibiting MIC values of 2 mg/L (susceptible) and 4 mg/L (resistant) by the ComASP^®^ cefiderocol microdilution panel and the reference BMD, respectively. Two major errors (MEs) were observed in two KPC-producing *K. pneumoniae* which both showed MIC values of 4 mg/L (resistant) and 2 mg/L (susceptible) by the ComASP^®^ cefiderocol microdilution panel and the reference BMD, respectively.

## 3. Discussion

Cefiderocol susceptibility testing represents a major challenge for clinical microbiology since reference microdilution requires ID-CAMHB, whose preparation is difficult to implement in the vast majority of laboratories. This technical requirement is also an obstacle to the inclusion of the cefiderocol test in commercial antimicrobial susceptibility panels. Lyophilized panels (Sensititre^TM^, ThermoFisher Scientific, Waltham, MA, USA) containing scalar concentrations of cefiderocol and a proprietary chelator in the wells bypassed the requirement for ID-CAMHB. First evaluations showed that Sensititre^TM^ panels were substantially equivalent to reference broth microdilution and received FDA clearance [18,19,20]. In August 2022, EUCAST evaluated the commercially available tests for cefiderocol susceptibility reporting that all presented problems related to accuracy, reproducibility, bias and/or for some, skipped wells. Consequently, Sensititre^TM^ panels have been withdrawn from the market. In such conditions, it is very complicated to find a solution to interpret the ATU results in laboratory routines. Considering the current limitations, EUCAST recommends starting testing cefiderocol with disc diffusion, which is predictive of susceptibility and resistance outside the ATU. Inside the ATU, alternative methods to resolve interpretative uncertainties are desirable (https://www.eucast.org/ast-of-bacteria/warnings, accessed on 5 January 2023). Recently, Liofilchem has launched the ComASP^®^ Cefiderocol microdilution panel, a two-test panel containing the dried antibiotic in 15 two-fold dilutions for the quantitative determination of the cefiderocol MIC against Gram-negative non-fastidious organisms, such as Enterobacterales, *P. aeruginosa*, *Acinetobacter* spp. and *S. maltophilia* (https://www.liofilchem.com, accessed on 5 January 2023). Unlike previous Sensititre^TM^ panels, this system uses ID-CAMHB, thus representing the first commercial method that mirrors the reference BMD.

Herein, we evaluated the in vitro activity of cefiderocol on a collection of difficult-to-treat bacterial isolates collected in a four-year period in different Italian hospitals following an algorithm based on disc diffusion and the ComASP^®^ cefiderocol panel. Following the EUCAST guidelines, disc diffusion allowed us to determine cefiderocol susceptibility (susceptible or resistant) in 78.6%, 88.1%, 85.4% and 100% of Enterobacterales, *P. aeruginosa*, *A. baumannii* and *S. maltophilia* isolates tested, respectively. Thus, excluding *S. malthophilia*, the rate of uninterpretable results ranged from 12 to 22% depending on the bacterial species. Moreover, a higher percentage of ATU results was found in the subset of ceftazidime/avibactam-resistant KPC producer isolates than in the ceftazidime/avibactam-susceptible isolates (43% vs. 11.4%). Similarly, a higher rate of cefiderocol resistance emerged in the same ceftazidime/avibactam-resistant group. These data were consistent with recent evidence highlighting the involvement of mutations in the omega loop of the KPC enzyme in determining co-resistance toward ceftazidime/avibactam and cefiderocol [11,21,22]. The ComASP^®^ cefiderocol microdilution panel was shown to be a valid method to determine cefiderocol MIC on isolates for which the disc diffusion results were uninterpretable. In fact, among the 50 isolates tested, three categorical errors were obtained, of which, only one was a VME, exhibiting 94% and 84% of overall categorical and essential agreement, respectively. Both MEs and VME were observed for MIC values nearly close to the resistance breakpoint (2 mg/L).

Overall, 20.5% of the carbapenemase-producing Enterobacterales that achieved ATU results by the disc diffusion method tested resistant by both the ComASP^®^ cefiderocol microdilution panel and the reference BMD. Conversely, all VIM-producing *P. aeruginosa* showed MIC values in the susceptible range (≤2 mg/L). Lastly, six out of seven (85.7%) A. baumannii tested by the ComASP^®^ cefiderocol microdilution panel and the reference BMD were resistant to cefiderocol according to EUCAST PK/PD breakpoints (susceptible ≤2 mg/L, resistant >2 mg/L) suggesting that zone diameters <17 mm were not unequivocally suggestive of resistance. Overall, among resistant isolates, cefiderocol MIC values ranged from 4 to 16 mg/L in both Enterobacterales and *A. baumannii* isolates.

Although surveillance studies have reported potent activity with low MIC90 and MIC50 values, even on MDR isolates, reports of cefiderocol resistance are steadily increasing [11,12,13]. The combination of several mechanisms, including β-lactamases and mutations affecting siderophore receptor expression or function, appears to be the main cause of reduced susceptibility or resistance to cefiderocol [14]. Considering both the disc diffusion and MIC determination methods, our study showed resistance rates to cefiderocol of 4.3%, 61.4%, 47.4% and 13% in ceftazidime/avibactam-susceptible KPC-producing Enterobacterales, ceftazidime/avibactam-resistant KPC-producing *K. pneumoniae*, NDM-producing Enterobacterales and carbapenemase-producing *A. baumannii* isolates, respectively. The high rates of resistance in ceftazidime/avibactam-resistant KPC-producing *K. pneumoniae* are already known, as well as among NDM producers [11,12,14,23]. A systematic review by Wang et al. showed that MIC values for NDM-producing Enterobacterales isolates were significantly higher than those harboring other β-lactamase genes, with a cefiderocol susceptibility rate of 83.4% [10]. In accordance with published data [10,24], our results showed that cefiderocol is highly active against VIM-producing Enterobacterales, carbapenemase-resistant *P. aeruginosa* and *S. maltophilia*, reaching 100% susceptibility. We should acknowledge that the results are limited to the Italian Piedmont epidemiology, which may be close to the epidemiology of other Italian regions and Western European countries.

## 4. Materials and Methods

### 4.1. Study Design

We investigated the in vitro activity of cefiderocol on a collection of 286 MDR Gram-negative clinical isolates collected from various clinical specimens (blood, rectal swabs, urine, respiratory samples) of patients admitted at six Italian Piedmont hospitals in the period 2019–2022. Disc diffusion was used as first-line antimicrobial susceptibility method. Subsequently, Enterobacterales and *P. aeruginosa* isolates that presented zone diameters within ATU (18–22 mm and 14–22 mm, respectively) were tested in parallel by the ComASP^®^ cefiderocol microdilution panel and the reference BMD. Likewise, *A. baumannii* and *S. maltophilia* that presented zone diameters <17 mm and <20 mm, respectively, were tested in parallel by the ComASP^®^ cefiderocol microdilution panel and the reference BMD.

### 4.2. Bacterial Isolates

Species identification was carried out by matrix-assisted laser desorption ionization-time of flight mass spectrometry (MALDI-TOF MS) (Bruker DALTONIK GmbH, Bremen, Germany). The detection of carbapenemase genes was carried out using an Xpert Carba-R assay (Cepheid, Sunnyvale, CA, USA) and an Amplex eazyplex^®^ SuperBug Acineto (AmplexDiagnostics GmbH, Gars am Inn, Germany) in Enterobacterales and *A. baumannii* isolates, respectively [25,26]. *P. aeruginosa* was investigated for metallo-β-lactamase production using E-test^®^ (BioMérieux, Marcy l’Etoile, France), and in the case of positivity, by an Xpert Carba-R assay [23].

### 4.3. Antimicrobial Susceptibility Testing

Antimicrobial susceptibility to main antimicrobials was tested using panel NMDR on a Microscan WalkAway automated microdilution system (Beckman Coulter, Nyon, Switzerland). Cefiderocol testing was performed using disc diffusion according to EUCAST guidelines. Cation-adjusted Mueller–Hinton agar (Becton-Dickinson, Franklin Lakes, NJ, USA) and a cefiderocol 30 µg disc (Oxoid Ltd., Basingstoke, UK) were used. Sequentially, the ComASP^®^ cefiderocol microdilution panel and the reference BMD were used in parallel to test cefiderocol on isolates that presented ATU or non-interpretable results by disc diffusion. Reference broth microdilution was performed by using an ID-CAMHB broth as recommended by EUCAST guidelines and using 0.015–32 mg/L of cefiderocol as concentration range [27]. The ComASP^®^ cefiderocol microdilution panel was used according to the manufacturer’s instructions. Briefly, 0.4 mL of a 1:20 diluted 0.5 McF bacterial solution was added to the ID-CAMHB tube provided in the kit. Subsequently, 100 µL of the obtained solution was dispensed into each well of the panel, and it was incubated at 37 °C for 16–20 h in ambient air. At the end of the incubation period, the MIC values were visually read as the lowest concentration of antibiotic that completely inhibits organism growth. If trailing was observed the cefiderocol MIC was read as the first well in which growth was significantly reduced corresponding to a button of ≤1 mm or the presence of light haze/faint turbidity. For each test, the validity of the positive control was checked.

Antimicrobial susceptibility testing results were inspected and interpreted according to EUCAST recommendations by two operators. *E. coli* ATCC 25922 and *P. aeruginosa* ATCC 27853 were used as quality control strains on each experimental sitting, checking that the quality control results were within the specified ranges. Susceptibility data were interpreted according to EUCAST clinical breakpoints (v. 13.0 2023) [28].

### 4.4. Statistical Analysis

Cefiderocol MIC values obtained by the ComASP^®^ cefiderocol microdilution panel and the reference BMD were compared to delineate categorical agreement. Two different types of error were defined: VME and ME, indicating a variation of the interpretation from resistant to susceptible and from susceptible to resistant, respectively. Variations in scalar dilutions were analyzed in order to delineate EA, corresponding to the number of MIC values within one doubling the dilution of the reference results. Comparisons involving distributions of inhibition zone diameters were tested using the Mann–Whitney test (two-tailed). The X^2^ test was used to compare resistance rates between isolates subsets. For all tests, a *p* ≤ 0.05 was considered significant.

## 5. Conclusions

To the best of our knowledge, this is the first study evaluating a commercial version of the reference BMD for cefiderocol susceptibility testing. Our results, although obtained on a limited number of isolates, suggest that the combination of disc diffusion with the ComASP^®^ cefiderocol microdilution panel could be a viable solution to overcome the challenge of cefiderocol susceptibility testing in routine microbiology laboratories. Further studies on high numbers of multicenter bacterial isolates are needed to strengthen these findings and to evaluate the test in cases where the inhibition zones fall outside the areas of uncertainty.

## Figures and Tables

**Table 1 antibiotics-12-00604-t001:** Cefiderocol activity by disc diffusion against multidrug-resistant Gram-negative isolates included in the study (2019–2022).

		Susceptible			Resistant			Area of Uncertainty		
	No.	No., %	Range Zones	Median Zones	No., %	Range Zones	Median Zones	No., %	Range Zones	Median Zones
Enterobacterales										
KPC producer CZA-susceptible	70	60, 85.7%	23–28	26	2, 2.8%	15–16	-	8, 11.4%	19–22	20
KPC producer CZA-resistant	44	4, 9.1%	23–25	23	21, 47.7%	8–17	15	19, 43.2%	18–21	19
NDM producer	19	7, 36.8%	23–26	24	6, 31.6%	6–14	13	6, 31.6%	18–22	20
VIM producer	31	26, 83.9%	23–29	27	0	-	-	5, 16.1%	21–22	21
OXA-48-like producer	10	9, 90%	24–28	27				1, 10%	22	-
KPC/VIM co-producer	4	4, 100%	25–26	25	0	-	-	0	-	-
*P. aeruginosa*										
VIM producers	10	7, 70%	23–26	25	0	-	-	3, 30%	19–22	19
MβL non-producer	32	30, 93.7%	23–30	26	0	-	-	2, 6.2%	19–22	-
*A. baumannii*										
OXA-23 producers	46	41, 89.1%	18–25	21	-	-	-	5, 10.9%	6–15	6
OXA-23/NDM co-producer	2	0	-	-	-	-	-	2, 100%	6	-
*S. maltophilia*	18	18, 100%	28–31	29	0	-	-	0	-	-

Abbreviations: CZA, ceftazidime/avibactam; MβL, metallo-β-lactamase.

**Table 2 antibiotics-12-00604-t002:** MICs for cefiderocol using ComASP^®^ cefiderocol microdilution panel compared to reference broth microdilution on bacterial isolates showing cefiderocol inhibition zones inside the area of interpretative uncertainty.

Species	Genotype/Phenotype	Inhibition Zones mm	Reference Microdilution MIC mg/L	ComASP^®^ Cefiderocol Microdilution Panel MIC mg/L
*K. pneumoniae*	KPC-p CZA-R	18	16	4
*K. pneumoniae*	KPC-p CZA-R	18	4	**2**
*K. pneumoniae*	KPC-p CZA-R	19	8	16
*K. pneumoniae*	KPC-p CZA-R	19	8	8
*K. pneumoniae*	KPC-p CZA-R	19	2	1
*K. pneumoniae*	KPC-p CZA-R	19	2	4
*K. pneumoniae*	KPC-p CZA-R	19	2	0.5
*K. pneumoniae*	KPC-p CZA-R	19	2	1
*K. pneumoniae*	KPC-p CZA-R	19	0.25	0.5
*K. pneumoniae*	KPC-p CZA-R	19	1	2
*K. pneumoniae*	KPC-p CZA-R	20	2	1
*K. pneumoniae*	KPC-p CZA-R	20	0.5	1
*K. pneumoniae*	KPC-p CZA-R	20	1	1
*K. pneumoniae*	KPC-p CZA-R	20	1	2
*K. pneumoniae*	KPC-p CZA-R	20	0.5	0.125
*K. pneumoniae*	KPC-p CZA-R	20	4	8
*K. pneumoniae*	KPC-p CZA-R	20	0.5	1
*K. pneumoniae*	KPC-p CZA-R	20	1	2
*K. pneumoniae*	KPC-p CZA-R	21	2	0.5
*K. pneumoniae*	KPC-p CZA-S	19	8	4
*K. pneumoniae*	KPC-p CZA-S	19	1	2
*K. pneumoniae*	KPC-p CZA-S	20	2	4
*K. pneumoniae*	KPC-p CZA-S	20	2	1
*K. pneumoniae*	KPC-p CZA-S	20	1	2
*K. pneumoniae*	KPC-p CZA-S	20	0.5	0.5
*K. pneumoniae*	KPC-p CZA-S	22	0.25	0.5
*K. pneumoniae*	KPC-p CZA-S	22	0.25	0.125
*K. pneumoniae*	NDM-p	18	8	4
*K. pneumoniae*	NDM-p	18	4	8
*K. pneumoniae*	NDM-p	19	1	2
*K. pneumoniae*	NDM-p	21	2	1
*E. coli*	NDM-p	21	4	8
*K. pneumoniae*	NDM-p	22	0.5	1
*E. cloacae*	VIM-p	22	0.25	0.5
*E. cloacae*	VIM-p	22	0.12	1
*E. cloacae*	VIM-p	21	2	2
*E. aerogenes*	VIM-p	21	1	1
*M. morganii*	VIM-p	21	1	0.25
*P. aeruginosa*	VIM-p	19	1	0.5
*P. aeruginosa*	MβL-np	19	0.25	0.5
*P. aeruginosa*	VIM-p	20	2	2
*P. aeruginosa*	VIM-p	22	0.5	1
*P. aeruginosa*	MβL-np	22	0.06	0.25
*A. baumannii*	OXA-23-p	6	16	32
*A. baumannii*	OXA-23-p	6	8	4
*A. baumannii*	OXA-23-p	6	4	8
*A. baumannii*	OXA-23-p	10	4	4
*A. baumannii*	OXA-23-p	15	2	1
*A. baumannii*	OXA-23/NDM-p	6	8	32
*A. baumannii*	OXA-23/NDM-p	6	16	16

According to EUCAST breakpoints, MIC values >2 mg/L and ≤2 mg/L indicate resistance and susceptibility, respectively. Grey shading indicates categorical errors: bold characters indicate very major errors; underlined characters indicate minor errors. Abbreviations: MIC, minimal inhibitory concentration; KPC-p, KPC producer; CZA-R, ceftazidime/avibactam-resistant; CZA-S, ceftazidime/avibactam-susceptible; NDM-p, NDM producer; VIM-p, VIM producer; MβL-np, metallo-β-lactamase non-producer; OXA-23-p, OXA-23 producer; OXA-23/NDM-p, OXA-23/NDM co-producer.

## Data Availability

The dataset analyzed during the current study is available from the corresponding author upon reasonable request.

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
