# Peer review of "Disc Diffusion and ComASP® Cefiderocol Microdilution Panel to Overcome the Challenge of Cefiderocol Susceptibility Testing in Clinical Laboratory Routine"

_antibiotics, 2023, doi:10.3390/antibiotics12030604_

Round 1
Reviewer 1 Report
The paper presents an important problem, which is resistance to drugs of last resort - carbapenems. The authors undertook the important task of evaluating the susceptibility of carbapenem-resistant strains to cefiderocol. Noteworthy is the fact that the study groups are unequal, Enterobacterales dominate among the strains, which affects the statistical analysis, which is quite poor.
In addition, the results of the work are not presented in an interesting way, i.e. table 2 is illegible, too extensive, maybe it is worth making a list of MIC ranges (?), and include detailed data in supplementary materials. There are also no graphs that would organize the data and interested the reader, e.g. box-plot. At the moment, the presentation of data reduces the value of the results obtained, and thus the work.
Furthermore, the references for verses 54-59 are missing, and the in vitro phrases are not in italics. Some paragraphs are indented and some are not, which requires unification.
The selection of literature deserves praise from the authors, the literature from the last 5 years dominates, mainly from 2022. The work itself is very interesting, but it requires work on the presentation of very important results from the epidemiological point of view.
Reviewer 2 Report
Overall content is fine. However, if the author would like to discuss more about how to implement the results to other area with a such resistant rate similar to this research results.
Author Response
Overall content is fine. However, if the author would like to discuss more about how to implement the results to other area with a such resistant rate similar to this research results.
We thank the reviewer for this comment. Accordingly, we enriched the discussion session.
Reviewer 3 Report
Review:
Source of antibiotic discs for disc diffusion testing of cefiderocol should be mentioned
Some details of the comASP cefiderocol in view of the medium used, chelator etc can be added
Any advantage of the product than sensititre ?
Strain number 5, 7, 8……etc although are not changing the interpretation, there are much variations in the reported results as against reference microdilution method, kindly address the situation. (page 3, table 2)
